# Diverse synthesis of α-tertiary amines and tertiary alcohols via desymmetric reduction of malonic esters

Haichao Liu[1], Vincent Ho Man Lau [1], Pan Xu[1], Tsz Hin Chan[1] & Zhongxing Huang [1] ✉

Amines and alcohols with a fully substituted α-carbon are structures of great value in organic synthesis and drug discovery. While conventional methods towards these motifs often rely on enantioselective carbon-carbon or carbon-heteroatom bond formation reactions, a desymmetric method is developed here by selectively hydrosilylating one of the esters of easily accessible α-substituted α-amino- and -oxymalonic esters. The desymmetrization is enabled by a suite of dinuclear zinc catalysts with pipecolinol-derived tetradentate ligands and can accommodate a diverse panel of heteroatom substituents, including secondary amides, tertiary amines, and ethers of different sizes. The polyfunctionalized reduction products, in return, have provided expeditious approaches to enantioenriched nitrogen- and oxygen-containing molecules, including dipeptides, vitamin analogs, and natural metabolites.

Amines and alcohols with a tetrasubstituted α-carbon are prevalent structural fragments in bioactive natural metabolites, including alkaloids, polyketides, and terpenoids[1,2]. These fully substituted carbons can also benefit drug discovery by bringing enhanced structural diversity and lipophilicity while blocking undesired enzymatic degradation[3,4]. As a result, continuous efforts have been devoted to construct α-tertiary amines and tertiary alcohols, particularly their polyfunctionalized derivatives like amino acids, hydroxyesters, and diols, in an efficient and stereoselective manner. Traditional methods to access these chiral motifs often rely on face-selective bond formation reactions of planar substrates and intermediates, including nucleophilic addition to ketones or imines[5–8], electrophilic oxidation or amination[9,10], substitution reactions of oxygen- or nitrogen-bearing nucleophiles[11,12], and rearrangement reactions[13] (Fig. 1a). Recent advances in transition metal and enzymatic catalysis have also allowed the asymmetric amination and oxidation of tertiary C–H bonds to generate these α-tetrasubstituted structures[14,15].

Recently, we have accomplished the reductive desymmetrization of all-carbon-[16,17] and halogen-substituted[18] malonic esters using a series of dinuclear zinc complexes with a tetradentate ligand. It is envisioned that the inclusion of amino- and oxymalonic esters in this desymmetrization paradigm[19,20] would provide an expeditious and diverse synthesis of enantioenriched α-tertiary amines and tertiary alcohols, as these diester substrates (**4** and **5**) can be rapidly and modularly prepared from inexpensive amino-, bromo-, and ketomalonic esters (**1**–**3**) via facile substitution and addition reactions (Fig. 1b). Nevertheless, there are additional challenges associated with these nitrogen-/oxygen-containing substrates. First, unlike inert alkyl and aryl groups, coordinating and/or acidic nitrogen/oxygen substituents can deactivate the bimetallic catalyst containing both a Lewis acidic and Brønsted basic zinc center[21]. Second, as the enantioselection of the desymmetrization boils down to the size difference between two α-substituents, a pair of carbon substituents of distinct sizes (e.g., phenyl vs methyl) were often prerequisite for a highly stereoselective hydrosilylation to quaternary stereocenters[16,17]. Comparably, a wide scope of amino- and oxymalonic esters would require catalysts to accommodate heteroatom substituents of diverse shapes and sizes, such as ethers, amides, and alkyl amines, to contrast different carbon sidechains (i.e., small carbon vs large heteroatom substituents and vice versa). Third, the synthetic application of the desymmetrization products hinges heavily on the derivatization potential of the nitrogen/oxygen substituents. As a result, the selection of protecting groups for amines and alcohols that are robust and can be removed under mild conditions is nontrivial.

---

[1]State Key Laboratory of Synthetic Chemistry, Department of Chemistry, The University of Hong Kong, Hong Kong, China. ✉e-mail: huangzx@hku.hk

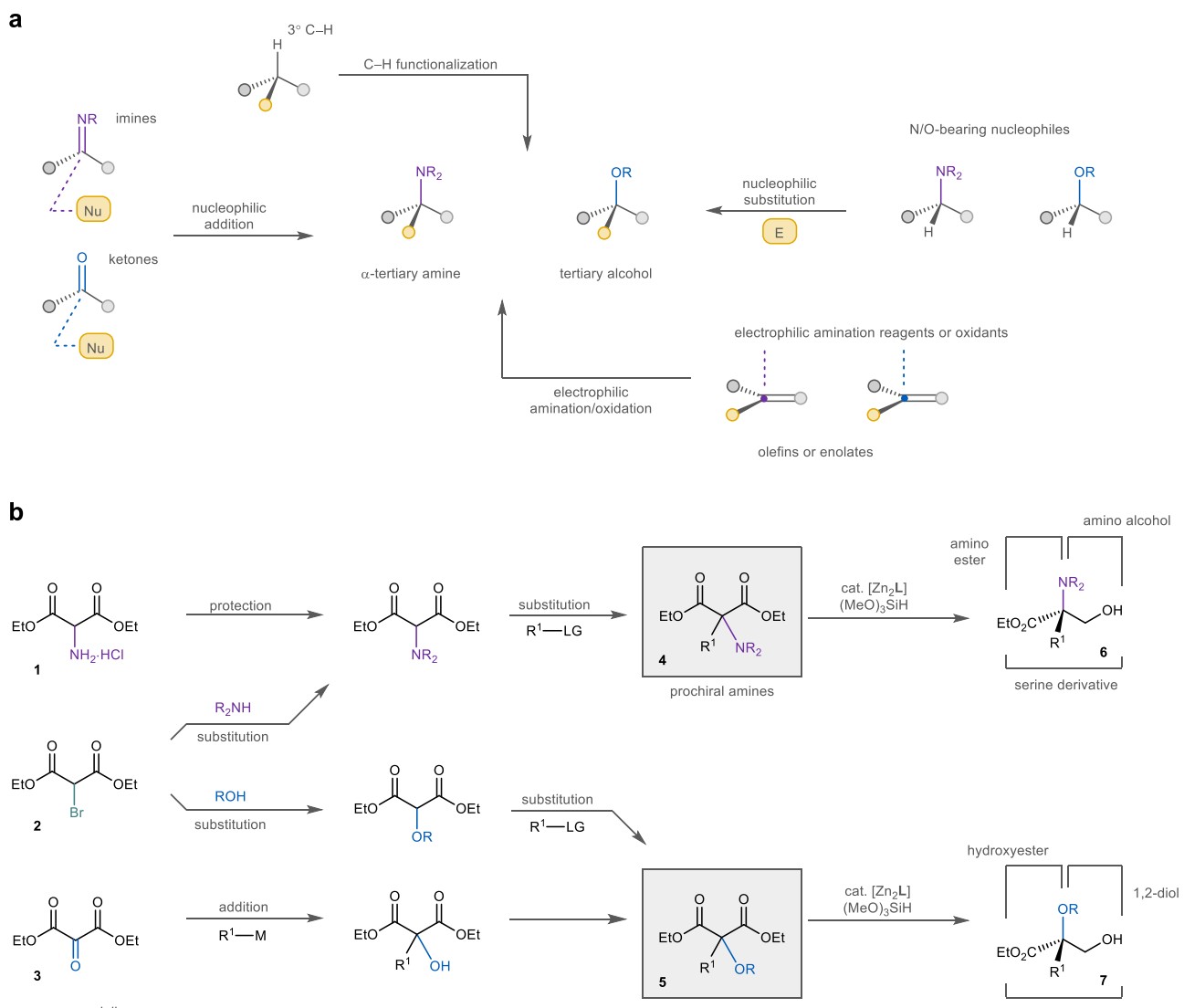

**Fig. 1 | Synthesis of chiral α-tertiary amines and tertiary alcohols. a** Synthesis of chiral α-tertiary amines and tertiary alcohols. Nu nucleophile, E electrophile. **b** Reductive desymmetrization of amino- and oxymalonic esters (this work). The prochiral amine and alcohol substrates of the desymmetrization can be easily and modularly prepared from commercially available starting materials **1**–**3**. M metal, LG leaving group.

In this work, we show that a suite of pipecolinol-derived tetradentate ligands can accommodate various types of nitrogen-/oxygen-substituted malonic esters to deliver the enantioenriched hydrosilylation products. While similar malonic esters have been desymmetrized before, these transformations were intramolecular and only chiral heterocycles were accessed in most cases[22–25]. In comparison, the reductive and intermolecular desymmetrization brings abundant application potential with the resulting β-hydroxyester motif and heteroatom substituents. For example, while the desymmetrization product of aminomalonate (**6**) can be viewed as highly functionalized amino alcohols/esters and serine derivatives, oxymalonic esters (**5**) can lead to oxygen-rich chiral molecules, such as diols and glyceric acids (**7**). Moreover, as the amino- and oxy-substituents do not participate in the hydrosilylation, a wide scope of them, including ethers, amines, and amides, is obtained. When coupled with a large collection of compatible sidechains (i.e., R[1]), the diversity of accessible structures via the desymmetrization is further expanded and facilitates the post-reduction derivatization to chiral nitrogen- and oxygen-containing molecules of higher complexity.

## Results and discussion
### Ligand identification
We initiated our exploration using aminomalonic ester equipped with a common benzoyl protecting group (Fig. 2a, **8**). Unfortunately, reactions with prolinol-based **L1** and **L2**, two privileged ligands for all-carbon-substituted malonic esters[22], returned no desymmetrization product. On the other hand, the aminomalonate shares a similar preference of tetradentate ligands with halomalonic esters[18], as pipecolinol ligands (**L3**–**L5**), particularly those with a pair of 3,5-disubstituted aryl groups (**L4** and **L5**), revived the reactivity of hydrosilylation. Meanwhile, the steric hindrance of the achiral *gem*-diaryl motif (i.e., Ar[1]) is equally important. Changing 1-naphathyl to a bare phenyl group (**L6**) largely inhibits the reduction, yet replacement with 3,5-di-*tert*-butylphenyl (**L7**) gives a similar level of enantiocontrol. Although *ortho*-substituted aryls (**L8** and **L9**) gave moderately better performance relative to fused naphthyls, a methylated 7-indolyl motif (**L10**) significantly improved both the yield and enantioselectivity.

At the same time, to devise a flexible desymmetric synthesis that caters for various carbon sidechains of malonic esters, we have identified and tested six representative categories of substrates (Fig. 2b).

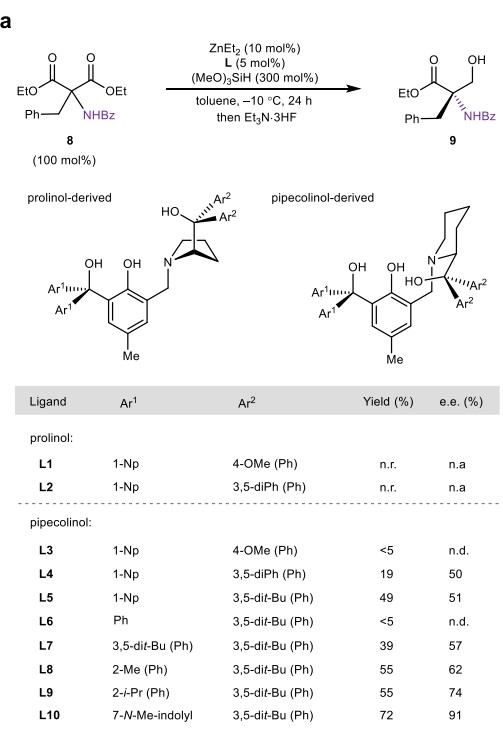

**a**

| Ligand | Ar$^1$ | Ar$^2$ | Yield (%) | e.e. (%) |
|--------|--------|--------|-----------|----------|
| **prolinol:** | | | | |
| **L1** | 1-Np | 4-OMe (Ph) | n.r. | n.a |
| **L2** | 1-Np | 3,5-diPh (Ph) | n.r. | n.a |
| **pipecolinol:** | | | | |
| **L3** | 1-Np | 4-OMe (Ph) | <5 | n.d. |
| **L4** | 1-Np | 3,5-diPh (Ph) | 19 | 50 |
| **L5** | 1-Np | 3,5-di$t$-Bu (Ph) | 49 | 51 |
| **L6** | Ph | 3,5-di$t$-Bu (Ph) | <5 | n.d. |
| **L7** | 3,5-di$t$-Bu (Ph) | 3,5-di$t$-Bu (Ph) | 39 | 57 |
| **L8** | 2-Me (Ph) | 3,5-di$t$-Bu (Ph) | 55 | 62 |
| **L9** | 2-$i$-Pr (Ph) | 3,5-di$t$-Bu (Ph) | 55 | 74 |
| **L10** | 7-$N$-Me-indolyl | 3,5-di$t$-Bu (Ph) | 72 | 91 |

**Fig. 2 | Optimization of reductive desymmetrization of amino- and oxymalonic esters. a** Optimization of ligand scaffold. Unless noted otherwise, the desymmetrization was run using **8** (0.1 mmol), (MeO)$_3$SiH (0.3 mmol), tetradentate ligand (5 mol%), and ZnEt$_2$ (10 mol%) in toluene at −10 °C for 24 h. The yields were determined using crude NMR spectra after work-up and e.e. values by chiral HPLC analysis of isolated product. Bz benzoyl, e.e. enantiomeric excess, n.r. no reduction, n.a. not available, n.d. not determined due to low yield. **b** Examination of six representative amino- and oxyamlonic esters. The absolute configuration of **9** was determined by comparison with a known $N$-Boc-protected α-amino acid after hydrolysis and protection. The absolute configuration of **10** was determined by comparison with a known compound after transesterification to methyl ester. The absolute configuration of **12** was determined by comparison with chiral citramalic acid after a sequence of derivatization (see Supplementary Fig. 255). The absolute configuration of **13** was determined by comparison with a known compound after deprotecting of the methyl ether. Bn benzyl.

Besides benzoyl protecting group (Cat. I), malonic esters with a dibenzylamine motif (Cat. II) were also included, as the bulky tertiary amine is expected to contrast sharply with small carbon substituents (e.g., C1–C3) for good enantioselectivity. Also due to the same reason, both malonic esters with a large $p$-methoxyphenyl ether[26–28] (Cat. IV) or a small methoxy group (Cat. V) were investigated. These categories were all successfully desymmetrized using the panel of 3,5-di-$tert$-butylphenyl-substituted pipecolinol ligands with slightly different achiral sidearms (**L7**, **L9**, and **L10**). In addition, cyclic amines (Cat. III) and ethers (Cat. VI) are also compatible, affording enantioenriched heterocycles embedded with a tetrasubstituted stereocenter. It is worth noting that the determined absolute configuration of these desymmetrization products (i.e., **9**, **10**, **12**, and **13**) supported the size difference between two substituents of malonic esters as the dominant factor of the enantioselection by the dinuclear zinc catalyst.

**Substrate scope**

The study of substrate scope evolves around these six classes of substrates. Benzoylamino motif was found to match alkyl chains of different lengths and shapes (**15–19**) to give structurally diverse α-tertiary amines stereoselectively (Fig. 3). Pendant functional groups, such as olefin (**20**), alkenyl bromide (**21**), acetal (**22**), cyclic ether (**23**), and thioether (**24**), are all compatible and expected to assist further derivatization of the chiral amines. Substituted aryl and heteroaryl moieties (**25–29**) are also tolerated. Particularly, aryl ester (**28**) and nitro motifs (**29**), two functional groups that were known to undergo hydrosilylation[29], remained intact during the desymmetrization, showcasing the high chemoselectivity of the zinc catalyst. The excellent functional group compatibility also allowed us to access polyfunctionalized derivatives of biomolecules, such as DOPA (**30**) and tryptophan (**31**). When a $tert$-butyl ester (**32**) or carbamate (**33**) was attached, enantioenriched homoglutamic acid and ornithine analogs can be generated and these molecules can be applied as chemical differentiated 1,6-diester and 1,4-diamine, respectively. It is worth noting that the desymmetric hydrosilylation can even operate on dipeptides (**34–36**), and zinc catalysts of opposite configuration can be used to access both isomers (**35** and **36**) from a chiral substrate in good diastereoselectivity. Meanwhile, compared with primary alkyl groups, the pipecolinol ligand gave a poor enantiocontrol for malonic esters with a secondary alkyl or aryl motif (**37–39**). In these cases, the prolinol counterpart of **L10** (**L11**) can improve the enantioselectivity, probably due to the smaller five-membered amino alcohol that provides a more spacious catalyst pocket. Intriguingly, the switch of ligand also alternated the configuration of the hydrosilylation product, indicating two distinctive modes of enantiocontrol between catalyst pockets derived from prolinols and pipecolinols.

Dibenzylamine-substituted malonic esters (Cat. II) can also accommodate various alkyl groups. However, the scope is limited to small-sized motifs, such as ethyl (**40**), allyl (**41**), and propargyl (**42**), owing to the difficult substitution of the bulky dialkylaminomalonate with large electrophiles (e.g., **43**). Cyclic amino groups, such as morpholine (**44**), piperidine (**45**), and piperazine (**46**), are suitable substituents, allowing the synthesis of enantioenriched amines containing these pharmaceutically relevant motifs[30,31]. On the other hand, the exploration of cyclic aminomalonic esters (Cat. III) as the desymmetrization substrates yielded a diverse panel of nitrogen heterocycles embedded with a tetrasubstituted carbon, including non-natural prolinol (**11** and **47**) and pipecolinol (**48**) derivatives. The generality of the desymmetric method was further showcased by the stereoselective synthesis of chiral lactams of different ring sizes (**49–52**). Notably, all

these heterocyclic products are polyfunctionalized in nature with a variety of functional groups that are expected to facilitate post-reduction derivatization to more complex structures.

The desymmetrization also proceeded on a diverse library of oxymalonic esters (Fig. 4a). Assorted alkyl groups, such as methyl (**12**), ethyl (**53**), allyl (**54**), and propargyl (**55**), match well with large

**Fig. 3 | Scope of aminomalonic esters.** Unless noted otherwise, the desymmetrization was run using aminomalonic ester (0.3 mmol), (MeO)₃SiH (0.9 mmol), tetradentate ligand (**L7**, **L9**, **L10**, or **L11**, 5 mol%), and ZnEt₂ (10 mol%) in toluene at −10 °C. The yields shown were isolated yields and e.e. values were determined by chiral HPLC analysis of isolated products. The d.r. values of **35** and **36** were determined by crude NMR. d.r. diastereomeric ratio.

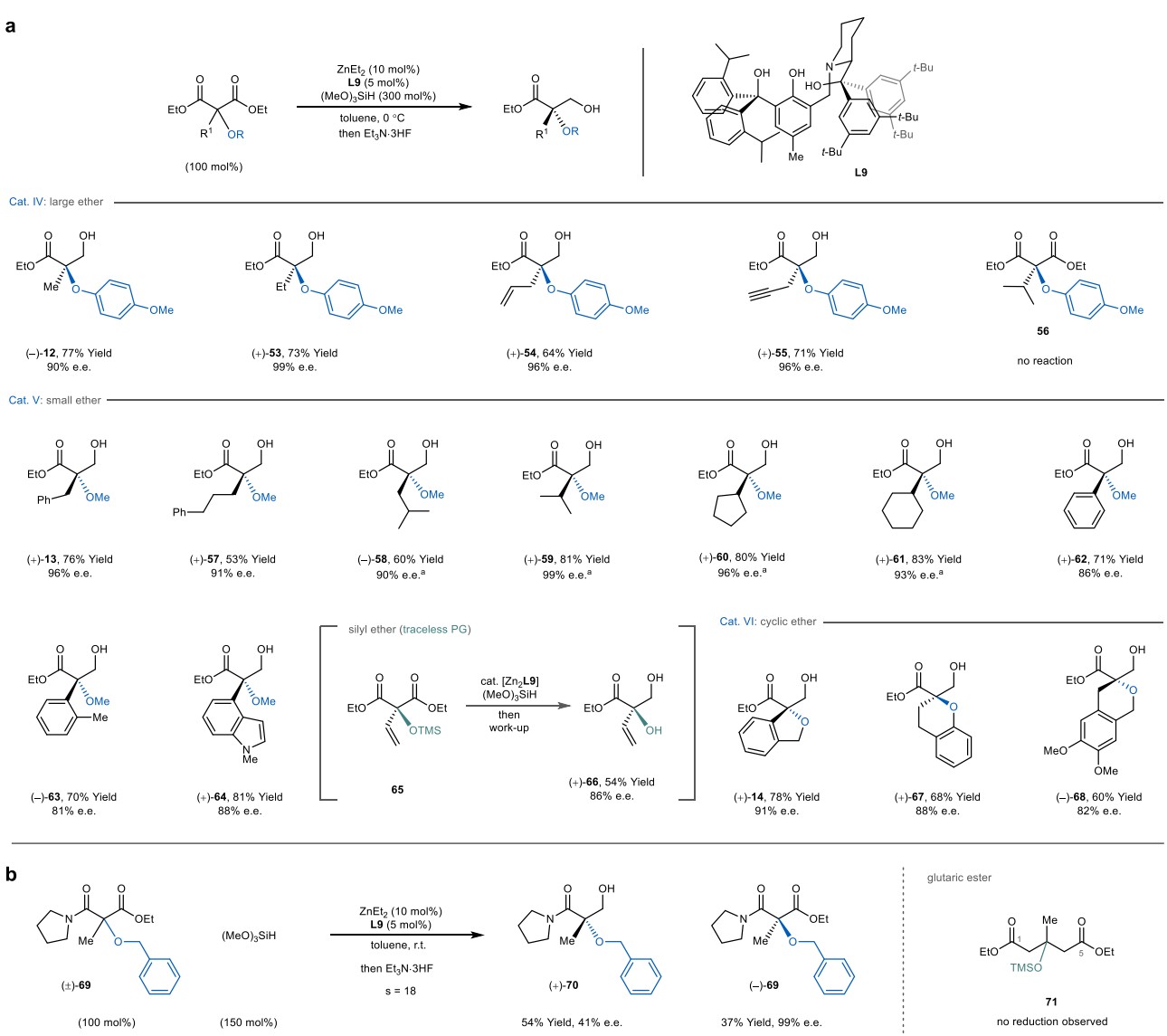

**Fig. 4 | Asymmetric hydrosilylation of oxymalonic esters. a** Scope of oxymalonic esters for the desymmetric hydrosilylation. Unless noted otherwise, the desymmetrization was run using oxymalonic ester (0.3 mmol), (MeO)₃SiH (0.9 mmol), **L9** (5 mol%), and ZnEt₂ (10 mol%) in toluene at 0 °C. The yields shown were isolated yields and the e.e. values were determined by chiral HPLC analysis of isolated products. TMS trimethylsilyl, PG protecting group. ᵃThe e.e. values were determined after an esterification with benzoyl chloride. **b** Kinetic resolution of oxymalonic esters and an attempted desymmetrization of diethyl glutarate. s, s factor of the kinetic resolution.

*p*-methoxyphenyl ether (Cat. IV) to give corresponding masked alcohols in excellent enantioselectivity. Nevertheless, the isopropyl-substituted substrate **56** failed to give any desymmetrization product. In comparison, both secondary (**13**, **57**, and **58**) and tertiary (**59**-**61**) alkyl motifs of large sizes are better paired with a methoxy group (Cat. V). The small size of the methyl ether also allowed the accommodation of large and rigid arenes (**62**–**64**), including those with an *ortho*-substituent (**63**) or fused ring (**64**). Intriguingly, we successfully applied silyl ether as a traceless protecting group in the desymmetrization of a vinyl-substituted malonic esters and the trimethylsilyl moiety was readily removed during the work-up to give glyceric ester **66**. Furthermore, the dinuclear zinc catalyst can house cyclic ethers (Cat. VI) of different shapes and desymmetrize the diesters to give chiral hydrobenzofuran (**14**), chromane (**67**), and isochromane (**68**). We have also attempted to extend the hydrosilylation reaction to asymmetric transformations other than desymmetrization, as well as substrates besides malonic esters (Fig. 4b). The kinetic resolution of oxymalonamate **69** proceeded smoothly to give a pair of enantioenriched starting material and β-hydroxyamide (**70**) with a moderate s

factor. On the other hand, however, glutaric esters (e.g., **71**) were unreactive, indicating the necessity of a 1,3-dicarbonyl motif for the dinuclear catalyst to operate on.

## Synthetic application
The application of desymmetrization to access bioactive molecules was subsequently examined. We first demonstrated that the methyl and *p*-methoxyphenyl ether can be smoothly deprotected using boron tribromide and ceric ammonium nitrate, respectively (Fig. 5a)[32]. Coupled with a cyanation and hydrolysis, the removal of the ether protecting group also provided a short synthesis of chiral citramalic acid (**74**) from desymmetrization product **12**. Although malic acid derivatives were shown to be accessible via the epoxidation and cyanation of the desymmetrization product from chloro/bromomalonic esters in our previous study[18], the differentiation between a methyl and halogen was poor, thus not able to afford citramalic acid with high enantioselectivity in principle. On the other hand, when reacting with an isothiocyanate, both the amine and ester of prolinol **11** participated to give the fused ring of macahydantoin B (**75**) rapidly (Fig. 5b)[33].

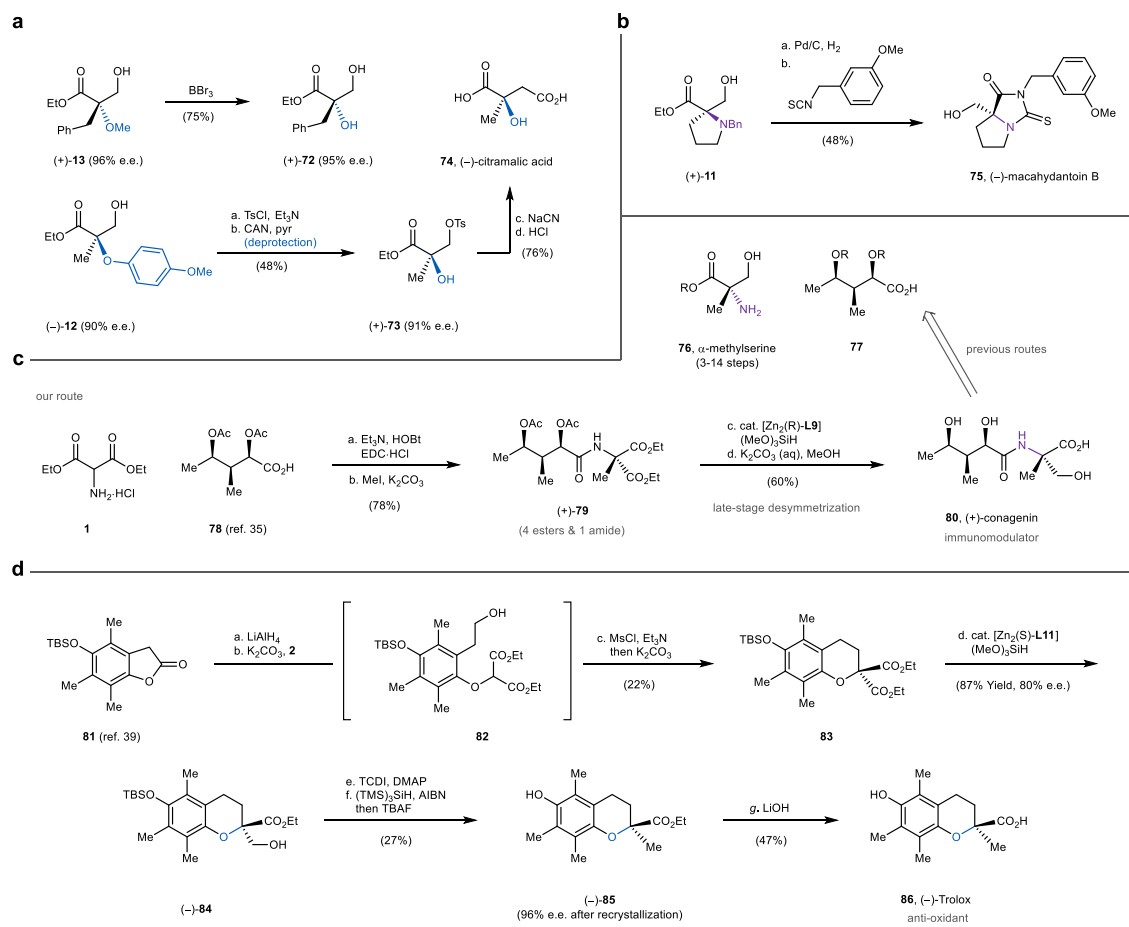

**Fig. 5 | Synthetic application of desymmetrization products. a** Deprotection of ethers and synthesis of citramalic acid. Ts toluenesulfonyl, CAN ceric ammonium nitrate, pyr pyridine. **b** Hydrogenative deprotection of benzylamine and synthesis of macahydantoin B. **c** A late-stage desymmetrization approach towards conagenin. Ac acetyl, HOBt 1-Hydroxybenzotriazole, EDC *N*-(3-dimethylaminopropyl)-*N*'-ethylcarbodiimide. **d** Synthesis of chiral Trolox via the desymmetrization of cyclic malonic ester. TBS *tert*-butyldimethylsilyl, Ms methanesulfonyl, TCDI thiocarbonyldiimidazole, DMAP 4-dimethylaminopyridine, AIBN azobisisobutyronitrile, TBAF tetra-*n*-butylammonium fluoride.

**Fig. 6 | Stereochemical consideration and proposal.** Our results indicate that ligands derived from prolinols and pipecolinols give hydrosilylation products of opposite configuration (**92** and **93**). The distinct stereoselectivity is proposed to arise from the different catalyst pockets. In the model of prolinol-derived ligand (**88** and **90**), the bulky diaryl motif of the prolinol would repel the large substituent of the malonic ester to the back and block the right front area in **90**. Therefore, the hydrosilylation takes place on the left ester. Meanwhile, the chair conformation of the pipecolinol would lower the barrier of the right front area in **91** significantly. While the larger substituent still points to the back, the hydrosilylation is mediated by the more exposed alkoxide on the right. $R^L$ larger substituent, $R^S$ smaller substituent.

We also devised a desymmetric approach toward conagenin (Fig. 5c). Reported accesses to this important immunomodulator often hinge on the amide bond formation between chiral acid **77** and α-methylserine derivative **76**, a building block nontrivial to synthesize stereoselectively[34]. In contrast, we employed a late-stage desymmetrization strategy, where the stereocenter on the methylserine residue was directly fashioned on a malonic ester (**79**) easily prepared from a known acid[35] (**78**) and commercially available aminomalonate **1**. Given that the hydrosilylation substrate (**79**) contains four esters and one amide, the approach clearly demonstrates the good site-, chemo-, and stereoselectivity of the dinuclear zinc catalyst. A similar malonate-based synthesis of Trolox[36–38] (**86**), a water-soluble antioxidant, was also successfully implemented (Fig. 5d). In this case, a cyclic and ethereal diester (**83**) was prepared from known lactone **81**[39] and bromomalonic ester **2**, followed by the desymmetrization and deoxygenation to generate the chiral chromane core (**85**).

During this study and our previous exploration of ligand scaffolds[16–18], prolinol- (e.g., **L11**), and pipecolinol-derived (e.g., **L9**) tetradentate ligands were found to give dominant hydrosilylation products of opposite configuration (Fig. 6, **92** and **93**). We propose that the switch of enantioselection results from the different conformation of the pyrrolidine and piperidine. As observed in similar complexes[40], the bulky diarylmethanol motif of the prolinol in **88** supposedly points away from the envelope plane and creates a high barrier for the catalyst pocket (i.e., the right front area of **90**). When the malonic ester adopts a bridged chelation[41,42] with the dinuclear zinc catalyst, the bis-aryl group would force the larger substituent of the substrate to the back of the catalyst, and the left ester in **90** is hydrosilylated selectively, presumably facilitated by the triarylmethoxide via a six-membered ring transition state. In the pipecolinol-based complex (**89**), however, the preferred diequatorial conformation of the six-membered ring significantly lowers the shield of the bis-aryl motif on the zinc center (i.e., the right front area of **91**). While R$^L$ is still directed to the back to reduce repulsion (**91**), the alkoxide on the right becomes more exposed to mediate the hydrosilylation.

In summary, we have devised an asymmetric synthesis of α-tetrasubstituted amines and alcohols via the desymmetrization of malonic esters. This approach takes advantage of the facile preparation of heteroatom-substituted malonic esters and bridges it with the high derivatization potential of the polyfunctionalized hydrosilylation products. The use of the pipecolinol-derived and tetradentate ligand scaffold with a pair of di-*tert*-butylphenyl groups allows the inclusion of substrates with a diverse array of amino and oxy functional groups. As a result, malonic ester-based syntheses have been enabled to access natural metabolites and bioactive molecules in a rapid and stereoselective fashion.

## Methods

### General procedure for the reductive desymmetrization

To an oven-dried 10-mL round bottom flask with a stir bar was added **L10** (45.0 mg, 0.05 mmol), then the flask was sealed with a rubber septum and evacuated/refilled with nitrogen for three times. Freshly distilled toluene (1 mL) was added to the flask via syringe and the mixture was stirred at room temperature for 5 min. Diethylzinc (0.10 mL, 1.0 M solution in hexane, 0.10 mmol) was added to the flask via syringe slowly. The resulting catalyst solution was stirred at room temperature for 30 min before use.

To a separate oven-dried 5-mL Schlenk tube with a stir bar was added aminomalonic ester (0.30 mmol, 100 mol%), sealed with a rubber septum, and evacuated/refilled with nitrogen for three times. Trimethoxysilane (110 mg, 0.90 mmol, 300 mol%) and freshly distilled toluene (2.7 mL) were added via syringe. The mixture was stirred at room temperature while 0.3 mL of catalyst solution was added via syringe. The reaction mixture was stirred at −10 °C for 36 h, monitored by thin-layer chromatography. After the starting material was consumed, 0.2 mL triethylamine trihydrofluoride was added dropwise to quench the reaction. The mixture was diluted with 5 mL diethyl ether and stirred for 30 min. Subsequently, the reaction mixture was filtered through a short pad of silica gel, eluted with diethyl ether slowly (as the remaining triethylamine trihydrofluoride reacts with silica gel to release heat). The filtrate was evaporated and purified by flash column chromatography (hexane/ethyl acetate) to yield the desymmetrization product.

## Data availability

The data supporting the findings of this study are available within the paper and its Supplementary Information.

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

## Acknowledgements
We thank the National Natural Science Foundation of China (no. 22171238, Z.H.), Research Grants Council of Hong Kong (no. 27301821, Z.H.), and the University of Hong Kong for financial support. We acknowledge funding support from the Laboratory for Synthetic Chemistry and Chemical Biology under the Health@InnoHK Program launched by the Innovation and Technology Commission, the Government of HKSAR. Ms. J. Yip and Ms. B. Yan are acknowledged for mass spectroscopy and NMR spectroscopy, respectively.

## Author contributions
Z.H. conceived and designed the project. H.L., V.H.M.L., P.X., T.H.C., and Z.H. carried out the experiments and analyzed the data. H.L. and Z.H. wrote the manuscript.

## Competing interests
The authors declare no competing interests.
