## [Peer Review File · Nature Communications]

REVIEWER COMMENTS

Reviewer #1 (Remarks to the Author):

The authors have reported a synthetic method for enantioenriched α -tertiary amines and alcohols via desymmetrizing hydrosilylation. A series of dinuclear zinc complexes with a tetradentate ligand has been applied for this reaction. Although enantioselective desymmetrization reactions have been well established (a recent review should be cited: doi.org/10.1016/j.tet.2022.132629), selective reduction of one of the symmetric diesters is not common. Through this method, diversified α -tertiary amines and alcohols were prepared from the corresponding diesters which are readily accessible. Six representative categories of substrates have been investigated to synthesize corresponding enantioenriched functionalized products in moderate to good yield with high ee. Additionally, this method could be applied for the synthesis of polyfunctionalized bioactive molecules, demonstrating the practicability of the method. However, possible intermediates responsible for asymmetric induction should be proposed.

Reviewer #2 (Remarks to the Author):

Reviewer report for “Desymmetric Synthesis of α -Tertiary Amines and Tertiary Alcohols” manuscript NCOMMS-22-14151

The manuscript describes a general method for the synthesis of enantioenriched α -tertiary amines and tertiary alcohols through a ligand-controlled reductive desymmetrization reaction of α -substituted α -amino- and oxy- malonic esters. I would like to recommend publishing the manuscript in Nature Communications, considering the outstanding merits of the work.

1. (Novel catalysts) The authors pointed out the unique challenges associated with these substrates because of the amino-/oxy- substituents' basicity and ability to coordinate to metal center, in comparison with their previous work on quaternary center and tertiary halogenated compounds (references 17 to 19). The authors were able to overcome the challenges by fine-tuning the structure of the catalysts (Scheme 2). Clearly, the study will lead to future advance in asymmetric catalysis.

2. (General applicability) Compared to existing methods, the desymmetrization approach is unified, as it separates the formation of fully substituted substrates from the enantioselective hydrosilylating step

(Figure 1 B). The rigidity of catalysts ensures that steric differentiation of the amino-/oxy-substituent and the geminal carbon substituent is achievable across a broad scope of substrates.

3. (Broad scope) Besides broadly applicable to a variety of substrate types, the reaction tolerates a diverse range of synthetically relevant functional groups (Figure 3, Figure 4). The products are obtained in high enantioselectivity. This is particularly important for the synthesis of complex molecules, as the authors successfully demonstrated in the synthetic applications of products (Figure 5).

In addition, I would like to offer a few suggestions for the authors to consider.

1. The title of the manuscript could be revised to better capture the many exciting characteristics of the study. The current title might not fully reflect the uniqueness and the outstanding results of the work (and could be mistaken as a review paper on the topic).

2. In the abstract (Line 10 to Line 11), it could be less confusing (as part of a standalone abstract) if the substrate types are specified as alpha-substituted alpha-amino- and oxy- malonic esters.

3. In the introduction the authors could further highlight the many competitive advantages of the new transformation, especially its adaptability to diverse substrate types and broad scope in comparison with existing methods for desymmetrization of amino-/oxy- malonic esters (examples in a recent review: *Tetrahedron*, 2022, 106-107, 132629).

4. In the introduction (Lines 46 to 49), the requirement to accommodate heteroatom substituents of diverse shapes and sizes to contrast with different carbon sidechains is not easily understandable without reading the subsequent parts. It might be useful to provide the readers with a balanced discussion on the challenges due to the added complexity to differentiate between hetero- and carbo- substituents, compared to two carbon sidechains (quaternary stereocenters, reference 17). In our study on a different system, we found that introducing an alpha-heteroatom improves the enantioselectivity of desymmetrization (*J. Am. Chem. Soc.* 2022, 144, 123).

5. In Figure 2 (also Figures 3, and 4), the absolute configurations of products are shown in such a way that compound 9 and compound 10 are opposite, and compound 12 and compound 13 are opposite. The stereochemistry of compound 9 has been assigned (Figure 3, footnote a). It would strengthen the hypothesis that the size difference between two alpha-substituents are determining factor (Line

47) if the absolute stereochemistry of the other type of compound (i.e., compounds 9 vs 10 or 12 vs 13) can be assigned.

6. Figure 3 part 1, in the bracket: the cyclopentyl group of compound 39, when serving as a substituent, is secondary. It could be ambiguous to refer it as a tertiary (3°) alkyl substituent.

7. Figure 5. Additional lines that separate the sections could make the boundaries clear.

8. Some of the references are incomplete (e.g., references 19, 20, missing journal names).

Reviewer #3 (Remarks to the Author):

Huang and coworkers have developed a dinuclear zinc-catalyzed desymmetrization reaction of amino- and oxymalonic esters by employing pipercolinol-derived tetradentate chiral ligands. This approach provides efficient access to enantioenriched α -tertiary amines and tertiary alcohols bearing quaternary stereocenters with high enantioselectivity. The combination of organozinc with a tetradentate ligand is crucial for this desymmetrization reaction; a modified ligand enabled the current desymmetric reaction of amino- and oxymalonic esters. Synthetic transformations of the products have been achieved towards enantioenriched nitrogen- and oxygen-containing molecules, including dipeptides, vitamin analogs, and natural metabolites in short steps, demonstrating the great practical value of this reaction. This is a beautiful expansion of their recent works in dinuclear zinc-catalyzed desymmetrization reactions. Nevertheless, considering the same chemistry has been well disclosed by the author's group (Nat. Chem. 2021, 634; J. Am. Chem. Soc. 2022, 144, 1951; J. Am. Chem. Soc. 2022, 144, 15, 6918) employing the similar catalysts, the variation of substituents of malonate substrates for this desymmetric reduction reaction could not guarantee its publication on Nat Commun. Therefore, this review is not willing to recommend its publication. Other comments: 1) Kinetic resolution has been realized in their previous work (Nat. Chem. 2021, 634), could this strategy be possible in the current reaction? 2) Amino- and oxymalonic esters were employed as the starting materials in this work, could the desymmetrization of dimethyl glutarate derived substrates be achieved under this kind of catalytic system? 3) NMR spectra for compounds S16, S24, S38, S47 contains impurities.

香 港 大 學
THE UNIVERSITY OF HONG KONG

Zhongxing Huang, Ph.D.
Assistant Professor
Department of Chemistry
University of Hong Kong
Hong Kong, China
Email: huangzx@hku.hk
Phone: (852)-5108-3085

July 1, 2022

Dear Reviewers,

Hope you and your family are doing well. We would like to first express our gratitude to you all for spending time reviewing and evaluating our manuscript. We are deeply encouraged by your kind comments, and all your suggestion and recommendation have helped improve our work greatly. Here, we invite you to assess our responses and your input will be highly appreciated.

Re: Reviewer 1

(1) **Comment:** ‘a recent review should be cited: doi.org/10.1016/j.tet.2022.132629’

Response: We appreciate the reviewer’s suggestion. The review has been added as ref. 17.

(2) **Comment:** ‘However, possible intermediates responsible for asymmetric induction should be proposed.’

Response: We agree with the reviewer and have added a separate figure (Figure 6) and paragraph to discuss the proposed stereochemical model of the desymmetrization. Particular attention has been dedicated to the intriguing switch of enantioselectivity between prolinol- and pipercolinol-derived tetradentate ligands.

Text added to manuscript:

‘During this study and our previous exploration of ligand scaffolds²²⁻²⁴, prolinol- (e.g., **L11**) and pipercolinol-derived (e.g., **L9**) tetradentate ligands were found to give dominant hydrosilylation products of opposite configuration (Figure 6, **92** and **93**). We propose that the switch of enantioselection results from the different conformation of the pyrrolidine and piperidine. As observed in similar complexes⁴⁰, the bulky diarylmethanol motif of the prolinol in **88** supposedly points away from the envelope plane and creates a high barrier for the catalyst pocket (i.e., the right front area of **90**). When the malonic ester adopts a bridged chelation⁴¹⁻⁴² with the dinuclear zinc catalyst, the bis-aryl group would force the larger substituent of the substrate to the back of the catalyst, and the left ester in **90** is hydrosilylated selectively, presumably facilitated by the triarylmethoxide via a six-membered ring transition state. In the pipercolinol-based complex (**89**), however, the preferred diequatorial conformation of the six-membered ring significantly lowers the shield of the bis-aryl motif on the zinc center (i.e., the right front area of **91**). While R^L is still directed to the back to reduce repulsion (**91**), the alkoxide on the right becomes more exposed to mediate the hydrosilylation.’

Figure added to manuscript:

Fig. 6| Stereochemical consideration and proposal.

Re: Reviewer 2

(1) **Comment:** ‘The title of the manuscript could be revised to better capture the many exciting characteristics of the study.’

Response: We thank the reviewer for this nice suggestion. The title has since been expanded as ‘*Modular and Diverse Synthesis of α -Tertiary Amines and Tertiary Alcohols via Desymmetric Reduction of Malonic Esters*’. We believe the new title reflects the easy access to diverse nitrogen-/oxygen-containing molecules provided by the desymmetrization and briefly covers the nature of the transformation.

(2) **Comment:** ‘In the abstract (Line 10 to Line 11), it could be less confusing (as part of a standalone abstract) if the substrate types are specified as alpha-substituted alpha-amino- and oxy- malonic esters.’

Response: The change has been made accordingly in the abstract.

(3) **Comment:** ‘In the introduction the authors could further highlight the many competitive advantages of the new transformation, especially its adaptability to diverse substrate types and broad scope in comparison with existing methods for desymmetrization of amino-/oxy- malonic esters’

Response: We thank the reviewer for this suggestion. We have since rewritten the corresponding part to compare our method with previous intramolecular transformation involving heteroatom-substituted malonic esters. These previous reports were also added as new references (ref. 18-21) for readers’ information.

Modified text in the manuscript:

‘While heteroatom-substituted malonic esters have been desymmetrized before, these transformations were intramolecular and only chiral heterocycles were accessed in most cases.¹⁸⁻²¹ In comparison, the reductive and intermolecular desymmetrization would bring abundant application potential with the resulting β -hydroxyester motif and heteroatom substituents. For example, while the desymmetrization product of aminomalonate (**6**) can be viewed as highly functionalized amino alcohols/esters and serine derivatives, oxymalonic esters (**5**) can lead to oxygen-rich chiral

molecules, such as diols and glyceric acids (7). Moreover, as the amino- and oxy-substituents do not participate in the hydrosilylation, a wide scope of them, including ethers, amines, and amides, is expected.'

(4) **Comment:** 'It might be useful to provide the readers with a balanced discussion on the challenges due to the added complexity to differentiate between hetero- and carbo- substituents, compared to two carbon sidechains'

Response: We agree with the reviewer. This part of text has been rewritten in a point-by-point fashion to illustrate these additional challenges compared with our previous work.

Modified text in the manuscript:

'First, unlike inert alkyl and aryl groups, coordinating and/or acidic nitrogen/oxygen substituents can deactivate the bimetallic catalyst containing both a Lewis acidic and Brønsted basic zinc center²⁵. Second, as the enantioselection of the desymmetrization boils down to the size difference between two α -substituents, a pair of carbon substituents of distinct sizes (e.g., phenyl vs methyl) were often prerequisite for a highly stereoselective hydrosilylation to quaternary stereocenters²²⁻²³. Comparably, a wide scope of amino- and oxymalonic esters would require catalysts to accommodate heteroatom substituents of diverse shapes and sizes, such as ethers, amides, and alkyl amines, to contrast different carbon sidechains (i.e., small carbon vs large heteroatom substituents and *vice versa*). Third, the synthetic application of the desymmetrization products hinges heavily on the derivatization potential of the nitrogen/oxygen substituents. As a result, the selection of protecting groups for amines and alcohols that are robust and can be removed under mild conditions is non-trivial.'

(5) **Comment:** 'It would strengthen the hypothesis that the size difference between two alpha-substituents are determining factor (Line 47) if the absolute stereochemistry of the other type of compound (i.e., compounds 9 vs 10 or 12 vs 13) can be assigned.'

Response: Following the reviewer's suggestion, we have confirmed the absolute configuration of 9, 10, 12, and 13 using the methods shown below. A footnote was added to Figure 2 to describe these configuration and determination methods. The hypothesis of the size difference as the determining factor of enantioselection was also noted in the main text.

Footnote added to Figure 2:

‘The absolute configuration of **9** was determined by comparison with a known *N*-Boc-protected α -amino acid after hydrolysis and protection. The absolute configuration of **10** was determined by comparison with a known compound after transesterification to methyl ester. The absolute configuration of **12** was determined by comparison with chiral citramalic acid (*vide infra*, Figure 5A) after a sequence of derivatization. The absolute configuration of **13** was determined by comparison with a known compound after deprotecting of the methyl ether.’

Text added to the manuscript:

‘It is worth noting that the determined absolute configuration of these desymmetrization products (i.e., **9**, **10**, **12**, and **13**) supported the size difference between two substituents of malonic esters as the dominant factor of the enantioselection by the dinuclear zinc catalyst.’

(6) **Comment:** ‘Figure 3 part 1, in the bracket: the cyclopentyl group of compound 39, when serving as a substituent, is secondary. It could be ambiguous to refer it as a tertiary (3°) alkyl substituent.’

Response: We thank the reviewer for pointing this out. These groups are now referred to as ‘secondary alkyl substituents’ in the text and figure.

(7) **Comment:** ‘Figure 5. Additional lines that separate the sections could make the boundaries clear.’

Response: Lines have been added for clarity.

(8) **Comment:** ‘Some of the references are incomplete’

Response: We thank the reviewer for pointing this out. The reference list has since been checked thoroughly and incomplete entries were corrected.

Re: Reviewer 3

(1) **Comment:** ‘This is a beautiful expansion of their recent works in dinuclear zinc-catalyzed desymmetrization reactions. Nevertheless, considering the same chemistry has been well disclosed by the author’s group employing the similar catalysts, the variation of substituents of malonate substrates for this desymmetric reduction reaction could not guarantee its publication on Nat Commun.’

Response:

We thank the reviewer for the kind word and totally understand the concern here. While the reductive desymmetrization of assorted malonic esters has been a collective effort by our group, we feel that the inclusion of nitrogen-/oxygen-substituted substrates is perhaps the most exciting advances of this reaction paradigm. Compared with our previous synthesis of all-carbon quaternary stereocenters and tertiary alkyl halides, the application potential of enantioenriched amine and alcohol products here are more immense and diverse, given the prevalence of related chiral motifs in bioactive compounds. The challenges are unique as well. Nitrogen/oxygen functional groups are not only more labile than carbon and halide moieties but can also adopt different shapes depending on their substituents.

To tackle these challenges in a comprehensive way, our study identified six key categories of malonic esters with four slightly distinct ligands that can desymmetrize each of these substrates in good yield and enantioselectivity. These categories cover both cyclic nitrogen/oxygen heterocycles and acyclic α -tertiary amine/alcohol derivatives. Particularly, carbon side chains of varied sizes can all be accommodated with high enantiocontrol when accompanied with either a big or small nitrogen/oxygen motif.

The suite of catalysts and the chemoselective hydrosilylation, in return, afforded polyfunctionalized structures that well exceed our previous desymmetrization methods in diversity and complexity. These structural motifs include a) enantioenriched derivatives of biomolecules like DOPA, tryptophan, and dipeptides; b) chiral heterocycles, such as morpholine, piperidine, piperazine, prolinol, pipercolinol, lactam, hydrobenzofuran, chromane, and isochromane; c) molecules containing a pair of the same functional groups that are chemically differentiated, such as 1,6-diester

(homoglutamic ester) and 1,4-diamine (ornithine analog). More importantly, when incorporated in a synthetic route towards nitrogen-/oxygen-containing complex molecules, the hydrosilylation can provide novel and expeditious approaches based on malonic esters. As a demonstration of these potential new approaches, quick accesses to four bioactive compounds, citramalic acid, macahydantoin B, conagenin, and Trolox, were accomplished in the manuscript.

We truly hope the unique challenges and reward of these heteroatom-substituted substrates, the highly flexible and compatible nature of the catalysts, and the new routes towards diverse nitrogen-/oxygen-containing molecules here could alleviate the reviewer's concern. With a high-quality and multi-disciplinary platform like Nat. Commun., we feel the study would attract not only researchers in synthesis and catalysis but also practitioners in medicinal chemistry, chemical biology, and pharmaceutical industry.

(2) **Comment:** 'Kinetic resolution has been realized in their previous work (Nat. Chem. 2021, 634), could this strategy be possible in the current reaction?'

Response: We appreciate the reviewer's suggestion. During the revision, a malonamic ester (**69**), the half-amide analog of malonic ester, was examined for the kinetic resolution. To our delight, enantioenriched starting material and hydrosilylation product were obtained with a moderate *s* factor. The result of the kinetic resolution is now added to the manuscript.

Figure added to the manuscript (Figure 4B):

Description added to the text:

'We have also attempted to extend the hydrosilylation reaction to asymmetric transformations other than desymmetrization, as well as substrates besides malonic esters (Figure 4B). The kinetic resolution of oxymalonamate **69** proceeded smoothly to give a pair of enantioenriched starting material and β -hydroxyamide (**70**) with a moderate *s* factor.'

(3) **Comment:** 'Amino- and oxymalonic esters were employed as the starting materials in this work, could the desymmetrization of dimethyl glutarate derived substrates be achieved under this kind of catalytic system?'

Response: Following the reviewer's recommendation, we have prepared a silyl ether-substituted glutaric ester (**71**) and tested its reactivity. Unfortunately, the diester didn't show any reactivity under the hydrosilylation conditions, indicating that 1,3-dicarbonyl motif, as in malonic esters, was essential for dinuclear zinc catalyst to proceed. Nevertheless, this failed reaction has been included in the manuscript for the readers' information.

Figure added to the manuscript (Figure 4B):

Description added to the text:

‘On the other hand, however, glutaric esters (e.g., **71**) were unreactive, indicating the necessity of a 1,3-dicarbonyl motif for the dinuclear catalyst to operate on.’

(4) **Comment:** ‘NMR spectra for compounds S16, S24, S38, S47 contains impurities.’

Response: We thank the reviewer for pointing this out. These spectra have been replaced with those of higher quality.

Again, we greatly appreciate your consideration and help in improving our manuscript. If you have any further input or there is anything else I can clarify, please feel free to let me know. Thanks!

Yours sincerely,

Zhongxing Huang

REVIEWERS' COMMENTS

Reviewer #1 (Remarks to the Author):

The manuscript was revised appropriately. It could be accepted for publication as it is.

Reviewer #2 (Remarks to the Author):

I recommend the publication of the revised manuscript.

The authors have addressed my comments submitted during the first round of review. In particular, the absolute configurations of products have supported the authors hypothesis that the steric effect is the determining factor in the stereocontrol.

An additional minor comment, structure 87 in the newly added figure 6 can be revised so that the bottom benzene ring is shown in its complete form. L9 and L11 share the same structure at this position.

Reviewer #3 (Remarks to the Author):

Considering the authors' response, the revisions, and comments from other reviewers, I now fully agree the publication of this manuscript as it stands.

香港大學
THE UNIVERSITY OF HONG KONG

Zhongxing Huang, Ph.D.
Assistant Professor
Department of Chemistry
University of Hong Kong
Hong Kong, China
Email: huangzx@hku.hk
Phone: (852)-5108-3085

July 30, 2022

Dear Reviewers and Dr. Malik,

We would like to thank you for your input and favorable consideration for our manuscript. We are highly encouraged and have made final revision on our manuscript. Besides one minor change to the drawing of one structure suggested by Reviewer 2, all the editorial comments have been addressed.

Re: Reviewer 2

(1) **Comment:** ‘An additional minor comment, structure **87** in the newly added figure 6 can be revised so that the bottom benzene ring is shown in its complete form. **L9** and **L11** share the same structure at this position.’

Response: We appreciate the reviewer’s suggestion. The structure has been modified accordingly.

Modified 87 in Figure 6:

Again, we greatly appreciate your consideration and help in improving our manuscript. If you have any further input or there is anything else I can clarify, please feel free to let me know. Thanks!

Yours sincerely,

Zhongxing Huang